# Remote Action Generation:
# Remote Control with Minimal Communication

## Abstract

We address the challenge of remote control where one or more actors, lacking direct reward access, are steered by a controller over a communication-constrained channel. The controller learns an optimal policy from observed rewards and communicates action guidance to the actors, which becomes demanding for large or continuous action spaces. To achieve rate-efficient communication throughout this interactive learning and control process, we introduce a novel framework leveraging *remote generation*. Instead of transmitting full action specifications, the controller sends minimal information, enabling the actors to *locally generate* actions by sampling from the controller's evolving target policy. This guided sampling is facilitated by an importance sampling approach. Concurrently, the actors use the received guidance as supervised learning data to learn the controller's policy. This actor-side learning improves their local sampling capabilities, progressively reducing future communication needs. Our solution, Guided Remote Action Sampling Policy (GRASP), demonstrates significant communication reduction, achieving an average 12-fold data reduction across all experiments (50-fold for continuous action spaces) compared to direct action transmission, and a 41-fold reduction compared to reward transmission.

## 1 Introduction

Reinforcement learning (RL) enables the solution of complex, sequential tasks through interaction with the environment alone. This is accomplished by identifying a sequence of actions that maximize the cumulative expected rewards. In this work, we consider a distributed learning scenario with two types of agents: a *controller* and an one or more *actors*. This setting is depicted in Figure 1. The actors observe the state of the environment, either fully or partially, and decide on an action; however, they do not have access to the reward signal. The controller observes both the state of the environment and the reward signal, but relies on the actors to take actions. The controller communicates with the actors over a rate-limited channel to help guide it toward the correct actions. We dub this problem *remote reinforcement learning (RRL) with a communication constraint.*

This scenario can model situations in which the controller has access to additional resources to evaluate or acquire the reward signal. For instance, in human-in-the-loop systems, the reward may need to be evaluated and provided by a human, which can cause delays (Knox & Stone, 2009; Daniel et al., 2014), or be learned from demonstrations (Abbeel & Ng, 2004; Schaal, 1996; Arora & Doshi, 2021). In other cases, the reward is available only through a large model that cannot be executed on the resource constrained actors, such as vision–language models that assess correctness of manipulation tasks (Rocamonde et al., 2024; Ma et al., 2023). Comparable challenges appear in active learning where reward signals are sparse or expensive to acquire (Krueger et al., 2020; Eberhard et al., 2024) and in AI-feedback models (Lee et al., 2024).

Multi-agent reinforcement learning (MARL) extends the traditional RL framework to multiple agents, where the agents collectively influence the environment's state. This scenario is particularly relevant to RRL because the reward is often tied to the overall system's performance; and thus, may not be directly accessible to each actor. Moreover, decentralized MARL suffers from a high degree of non-stationarity (Du & Ding, 2021; Wong et al., 2023). If each agent views others as part of the environment, the learning and policy updates by other

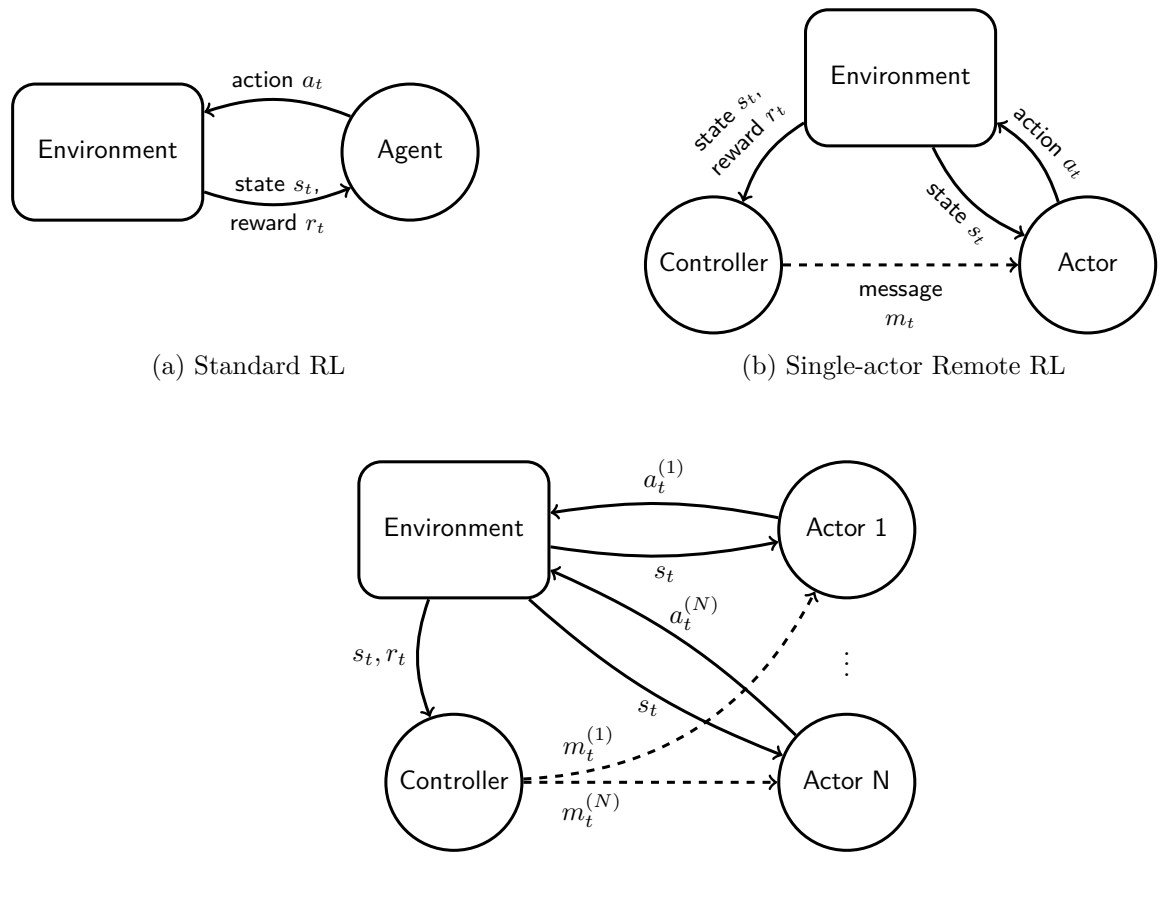

(a) Standard RL

(b) Single-actor Remote RL

(c) Multi-actor Remote RL

Figure 1: Reinforcement learning frameworks. (a) The standard RL loop where the agent observes state $s_t$ and reward $r_t$ from the environment and takes action $a_t$; (b) Remote RL where a controller with reward access sends messages $m_t$ to an actor that executes actions; (c) Multi-actor Remote RL where one controller coordinates multiple actors through separate communication channels.

agents alter the environment, rendering it highly non-stationary and challenging to learn from. To address this issue, a centralized-learning decentralized-execution approach is typically employed (Lowe et al., 2017). During training, this method involves centrally learning the policies of all agents using global information, thereby avoiding the non-stationarity problem. After training, these policies are fixed, ensuring that even though the agents execute them independently, the environment remains consistent for each agent. Multiple agents in MARL translate into multiple actors in RRL, while a single centralized controller is ideally suited to oversee the centralized training stage, enabling the actors to take correlated actions at each step.

Examples of multi-actor settings with centralized but costly reward evaluation include smart grid control, where the reward depends on global stability or optimal power flow computed centrally (Rossi et al., 2025), distributed packet routing in communication systems, where performance depends on system-wide latency or throughput metrics that only a central controller can compute (Al-Rawi et al., 2015), and traffic signal control, where rewards such as total waiting time or emissions require aggregation across the entire network rather than being observable at individual intersections (Li et al., 2024; Chu et al., 2020). In all these cases, the local actors observe only local state and cannot derive the global reward without central aggregation.

In this work, we propose a solution to RRL, called Guided Remote Action Sampling Policy (GRASP), where the controller learns the target policy using a standard RL algorithm. Trivially, the controller would sample an action from its target policy and send it directly. This would require communication rate growing with

the action space size, and some quantization for continuous action spaces. Crucially, the controller wants the agent to take *any* action as long as it follows the controller's target policy (i.e., the probability distribution over actions). Thus, our approach is different: the actor samples actions from its own proposal policy to create a list of candidate actions. The controller then uses importance sampling to choose one of them and communicates only the index of the desired action in the list. This significantly decreases the communication load and enables exact sampling from continuous distributions. Crucially, the actor simultaneously learns its own policy via imitation, using the guided actions as training data. While this actor policy is never directly executed, its improvement as a proposal distribution minimizes the communication required for the controller's guidance.

The remainder of the paper is organized as follows: Section 2 reviews related works, Section 3 introduces the necessary background. Section 4 mathematically defines the framework for RRL and details GRASP algorithm. Section 5 empirically evaluates the proposed approach, comparing it against other solutions. Finally, the paper concludes with a summary of findings and proposes potential future research directions.

The logarithms are base 2, $\mathbf{E}\left[\cdot\right]$ denotes expectation, $\mathbf{H}\left(P\right) \triangleq \mathbf{E}_{X \sim P}\left[\log p(x)\right]$ represents the entropy of a random variable distributed according to $P$, or differential entropy in the case of continuous random variables, and $\mathbf{D}_{KL}\left[P||Q\right] \triangleq \mathbf{E}_{x \sim P}\left[\log \frac{p(x)}{q(x)}\right]$ denotes the Kullback-Leibler divergence between distributions $P$ and $Q$.

## 2 Related Works

RL literature, particularly multi-agent scenarios, includes many connections to communications. Relevant works include federated RL (Nadiger et al., 2019; Jin et al., 2022), where multiple agents collaborate to learn a common policy while keeping data localized to each agent. This contrasts with RRL, where both the controller and the actors have access to the state and the actions. MARL with communication among agents is an extensively studied topic, where the agents exchange messages over a dedicated link (Foerster et al., 2016; Wang et al., 2020; Tung et al., 2021), to achieve a common goal. In these works, common reward is known to all the actors, unlike in our setting, where it is only accessible to a remote controller. RRL is orthogonal to these approaches, it can describe scenarios with or without communication between actors. Similarly, GRASP can be applied in both cases. Furthermore, in MARL with communication, the centralized training with decentralized execution paradigm is often employed, to which GRASP is particularly well-suited.

Beyond MARL, cloud-robotics and resource offloading literature also investigates splitting heavy computation between a centralized server and lightweight edge robots, accounting for bandwidth and latency constraints (Tahir & Parasuraman, 2025). Moreover, communication constraints have been studied in distributed learning and multi-armed bandit settings, focusing on reward or model compression to minimize regret (Hanna et al., 2022; Mitra et al., 2023; Salgia & Zhao, 2023). Unlike our problem, those works assume the actor sees the reward and sends it back. The work closest to ours is Pase et al. (2022), which studies sending actions over a communication-limited channel in a contextual multi-armed bandit problem. In contrast to our work, the states are independent across time, and the agents cannot learn the policy. The authors study the regret behavior for a certain class of policies, focusing on the asymptotic regime of infinitely many agents.

## 3 Background

### 3.1 Remote Generation

Given distributions $P$ and $Q$ at the encoder, what is the smallest average message length $m$ that allows the decoder to generate a sample from $P$, given that it only knows $Q$? This is the problem of *remote generation*. In GRASP, $P$ corresponds to the controller's policy $\pi_C$, while $Q$ corresponds to the actor's proposal policy $\pi_A$. The importance sampling procedure begins by generating a list of sample actions $(a_i)_{i=1}^{K}$, where $a_i \sim \pi_A$ for $i = 1, \ldots, K$. The decoder then selects one sample with probability proportional to the importance weight $w_i = \frac{p(x_i)}{q(x_i)}$ and communicates the index $i^*$ of the chosen sample to the decoder. When the list size $K$ is sufficiently large, on the order of $O(\exp \mathbf{D}_{KL}\left[P||Q\right])$, the chosen sample follows a distribution quantifiably close to $P$ (i.e., $\pi_C$). This procedure was proposed by Havasi et al. (2019), but it yields a suboptimal rate.

Therefore, in this work, we follow a slightly different algorithm (Theis & Yosri, 2022), which achieves the best known rate, upper-bounded by

$$\mathbf{D}_{KL}\left[P||Q\right] + \log\left(\mathbf{D}_{KL}\left[P||Q\right] + 1\right) + 4 \text{ bits.} \tag{1}$$

The exact procedure is described in Appendix C.

Without remote generation, sampling an action from $P$ at the controller and communicating it directly using lossless compression would require at least $\mathbf{H}\left[P\right] + \mathbf{D}_{KL}\left[P||Q\right]$ bits. In contrast, remote generation enables the actor to sample from $P$ by communicating only approximately $\mathbf{D}_{KL}\left[P||Q\right]$ bits. Notably, this approach allows sampling from a continuous distribution $P$ using a finite number of bits, provided that $\mathbf{D}_{KL}\left[P||Q\right] < \infty$.

## 3.2 Imitation learning

In the proposed solution to the RRL problem, actions need to be effectively communicated from the controller to the actor. As explored above, to facilitate this, we will use remote action generation, which enables the actor to take actions from the desired policy of the controller by using approximately $\mathbf{D}_{KL}\left[P||Q\right]$ bits, where $P$ represents the action probability distribution under the controller's policy in a given state, and $Q$ is a reference probability distribution known to both the controller and the actor. What should $Q$ be? One solution is to periodically transmit the controller's current policy to the actor and use it as the reference distribution $Q$. This method involves resending updates to account for the evolving policy as the controller learns. Since the policies are represented as neural networks, this approach requires periodically transmitting all the parameters, or their changes, which would be extremely costly from a communication perspective.

Alternatively, since the actor can observe the current state and receives a sample from the desired policy, it can learn the controller's policy—-a probability distribution over actions conditioned on the state—-in a supervised manner. This concept is known as *behavioral cloning* and is an application within imitation learning, a field focused on learning policies from demonstrations (Pomerleau, 1988; Torabi et al., 2018; Abbeel & Ng, 2004; Schaal, 1996; Arora & Doshi, 2021). Inverse RL (Arora & Doshi, 2021) offers another approach, where the objective is to recover the reward function from a set of state-action trajectories. While this approach can succeed in scenarios where behavioral cloning fails, it is also more complex, often requiring the solution of RL problems as a subroutine. A combination of these two approaches was proposed by Ho & Ermon (2016), where a policy is learned directly as if learning from rewards recovered through inverse RL, without explicitly solving the inverse problem. In our experiments, we found that behavioral cloning alone was sufficient for our purposes, and we provide a more thorough examination of this in Section 5.

## 4 Remote Reinforcement Learning (RRL)

In this section, we formally define the RRL problem in the multi-actor setting. The environment is modeled as a Markov decision process described by a tuple

$$M = (S, s_0, \{A^{(i)}\}_{i=1}^N, p_T, R, \gamma),$$

where $S$ is the set of states, $s_0$ is the initial state, $A^{(i)}$ is the action space of actor $i \in \{1, \ldots, N\}$, $p_T(s' \mid s, a_t^{(1)}, \ldots, a_t^{(N)}) : S \times A^{(1)} \times \cdots \times A^{(N)} \to \mathcal{P}(S)$ is the transition probability of moving to the next state $s'$ given the current state $s$ and the joint action $(a_t^{(1)}, \ldots, a_t^{(N)})$, $R(s_t, s_{t+1}, a_t^{(1)}, \ldots, a_t^{(N)}) : S^2 \times A^{(1)} \times \cdots \times A^{(N)} \to \mathcal{P}(\mathbb{R})$ is the reward function, and $\gamma \in [0, 1)$ is the discount factor (Sutton & Barto, 1998).

The objective is to find a set of policies $\{\pi^{(i)} : S \to \mathcal{P}(A^{(i)})\}_{i=1}^N$ that jointly maximize the expected discounted return:

$$\left(\pi^{(i)*}\right)_{i=1}^N = \operatorname*{arg\,max}_{\left(\pi^{(i)}\right)_{i=1}^N} \sum_{t=0}^{\infty} \gamma^t \mathbf{E}_{\substack{a_t^{(i)} \sim \pi^{(i)}(s_t) \\ s_{t+1} \sim p_T(s_{t+1}|s_t, a_t^{(1)}, \ldots, a_t^{(N)}) \\ r_t \sim R(s_t, s_{t+1}, a_t^{(1)}, \ldots, a_t^{(N)})}} \left[r_t\right]. \tag{2}$$

At each time step $t$, the current state $s_t$ is observed by the controller and all actors. The controller generates a set of variable-length messages

$$m_t^{(i)} = f^{(i)}(s_{[:t]}, r_{[:t-1]}), \quad i = 1, \ldots, N,$$

where each encoding function $f^{(i)} : S^t \times \mathbb{R}^{t-1} \to \{0,1\}^*$ produces a message intended for actor $i$. Each actor then selects an action based on its history and received messages:

$$a_t^{(i)} = g^{(i)}(s_{[:t]}, a_{[:t-1]}^{(i)}, m_{[:t]}^{(i)}),$$

where $g^{(i)} : S^t \times (A^{(i)})^{t-1} \times (\{0,1\}^*)^t \to A^{(i)}$.

## 4.1 Reward communication

If the controller is able to convey the reward signal to the actors through the communication channel, the actors would have all the necessary information to perform RL; that is, they could learn a policy that probabilistically maps states to actions to maximize the sum of future rewards. However, this approach encounters three primary limitations in RRL: parallelism, coordination and limited communication. Firstly, in MARL scenarios, where multiple actors jointly influence the same environment, simply conveying individual reward signals would result in a distributed training algorithm that struggles with action coordination. Secondly, to accelerate learning, multiple concurrent agents are often used to collect experiences independently (Mnih et al., 2016; Heess et al., 2017). In our framework, this corresponds to communicating with multiple actors, each interacting with the same but parallel environments. However, if the actors receive individual reward signals, they would develop distinct policies, failing to benefit from shared experiences. Lastly, the reward is usually a real number, and it may not be possible to represent it exactly with the finite number of bits dictated by the capacity of the communication channel between the controller and the actor. Thus, it would have to be quantized and compressed before being communicated. In practice, the reward is represented as a 32 or 64-bit floating point number; but, as we shall see later, this is many-fold larger than the communication rate required for remote RL. These limitations suggest that direct communication of the reward signal is not an efficient solution for RRL.

## 4.2 Action communication

Shifting the focus to the controller, which has full knowledge of the state and rewards, if it also had access to the actions, it could effectively run a RL algorithm locally to obtain the optimal policy. This would emulate the best possible performance of a centralized learning scenario, provided the controller can select and communicate the subsequent actions to the actors at each decision step. In scenarios involving small discrete action spaces, this method can result in smaller message sizes compared to conveying the reward signal (or a quantized version of it) to each of the actor. On the other hand, for continuous action spaces, one might initially think that communicating actions would face similar bandwidth limitations as with reward transmissions, given that actions in such spaces can assume an uncountably infinite number of values, necessitating some form of quantization.

However, crucially, the actors does not need to take a specific action from the controller's policy, but any sample from it would suffice. Let $P = \pi_C^i(\cdot \mid s)$ be the distribution of actions dictated by the controller's policy in a given state $s$ for actor $i$, while $Q = \pi_A^{(i)}(\cdot \mid s)$ represents the actor's belief about the policy in this state. From an information-theoretic perspective, the number of bits required to communicate a particular sample from $P$ (i.e., a specific action) is approximately $\mathbf{H}(P) + \mathbf{D}_{KL}[P||Q]$ bits—the entropy of the action plus the cost of using the 'wrong' distribution $Q$ to compress it.[1] Instead, by generating candidate samples from $Q$ and using $P$ only to select a single candidate via an importance-sampling-like criterion, the cost of communicating the index of the accepted sample can be reduced to approximately $\mathbf{D}_{KL}[P||Q]$ bits (Cuff,

---

[1]Typically, compressing a sample from a distribution $P$ requires $\mathbf{H}[P]$ bits. However, this assumes that both the encoder and decoder have access (perhaps implicitly) to the distribution $P$. This is not the case in RRL with action communication, as the policy $P$ is learned by the controller and not known by the actor. Therefore, communication must be performed using a code designed for another distribution, $Q$.

---

**Algorithm 1** GRASP Controller

---

**Require:** Initial controller policy parameters $\theta$, initial actor policy parameters $\{\phi^{(i)}\}_{i=1}^{N}$
1: **for** $epoch = 0$ **to** $T/\text{batch\_size}$ **do**
2:    **for** $step = 0$ **to** batch\_size **do**
3:       $t \leftarrow epoch \times \text{batch\_size} + step$
4:       $s_t \leftarrow$ observe state from environment
5:       **for** $i = 1$ **to** $N$ **do**
6:          $P^{(i)} \leftarrow$ controller's action distribution for actor $i$ : $(s_t, \theta)$
7:          $Q^{(i)} \leftarrow$ actor $i$'s action distribution : $(s_t, \phi^{(i)})$
8:          $a_t^{(i)}, m_t^{(i)} \leftarrow$ remote generation encoding$(P^{(i)}, Q^{(i)})$
9:          Send $m_t^{(i)}$ to actor $i$
10:      **end for**
11:      $r_t \leftarrow$ reward from environment
12:    **end for**
13:    $b \leftarrow epoch \times \text{batch\_size}$
14:    $e \leftarrow b + \text{batch\_size}$
15:    Update $\theta$ based on $s_{[b:e]}$, $\{a_{[b:e]}^{(i)}\}_{i=1}^{N}$, $r_{[b:e]}$ using online RL
16:    **for** $i = 1$ **to** $N$ **do**
17:       Update $\phi^{(i)}$ based on $s_{[b:e]}$, $a_{[b:e]}^{(i)}$ using supervised learning
18:    **end for**
19: **end for**

---

**Algorithm 2** GRASP Actor $i$

---

**Require:** Initial actor $i$ policy parameters $\phi^{(i)}$
1: **for** $epoch = 0$ **to** $T/\text{batch\_size}$ **do**
2:    **for** $step = 0$ **to** batch\_size **do**
3:       $t \leftarrow epoch \times \text{batch\_size} + step$
4:       $s_t \leftarrow$ observe state of the environment
5:       $Q^{(i)} \leftarrow$ actor $i$'s action distribution$(s_t, \phi^{(i)})$
6:       $m_t^{(i)} \leftarrow$ receive message from controller
7:       $a_t^{(i)} \leftarrow$ remote generation decoding$(m_t^{(i)}, Q^{(i)})$
8:       act in environment$(a_t^{(i)})$
9:    **end for**
10:   $b \leftarrow epoch \times \text{batch\_size}$
11:   $e \leftarrow b + \text{batch\_size}$
12:   Update $\phi^{(i)}$ based on $s_{[b:e]}$, $a_{[b:e]}^{(i)}$ using supervised learning
13: **end for**

---

2008; Li & El Gamal, 2018). This method of conveying random actions is particularly effective in systems with multiple parallel agents. By centrally processing all collected experiences, the controller can learn the most informed policy, benefiting from the experiences of all the actors in parallel. The controller can then enable each actor to take an action based on the most up-to-date policy in the next round. We call this approach GRASP.

The pseudocode for the proposed GRASP method is provided in Algorithm 1 for the controller and in Algorithm 2 for the actors. The controller maintains a copy of the actors' parameters; to employ remote generation, both parties (the encoder and decoder) need access to the common distribution $Q$. In GRASP, we employ actors' current policy conditioned on the current state as the common distribution. This policy is never enacted; that is, the actors' actions do not follow them directly, but are instead used solely to facilitate efficient communication of actions derived from the controller's policy via importance sampling. Additionally, the parameters of the actors' network are never explicitly communicated; they are updated based on the observed

actions and states, allowing them to evolve in lockstep between the actors and controller. In particular, to minimize the communication cost, we need to minimize the KL-divergence between the controller's policy $\pi_C$ and the actors' policies $\pi_A^{(i)}$, which corresponds to minimizing the empirical cross-entropy:

$$\underset{(\pi_A^{(i)})_{i=1}^N}{\arg\min} \mathbf{E}_s \left[ \mathbf{D}_{KL} \left[ \pi_C(\cdot|s) || \pi_A^{(1)}(\cdot|s) \times \cdots \times \pi_A^{(N)}(\cdot|s) \right] \right] \simeq \underset{(\pi_A^{(i)})_{i=1}^N}{\arg\min} \frac{1}{T} \sum_{t=1}^T \sum_{i=1}^N -\log \pi_A^{(i)}(a_t^{(i)}|s_t),$$

where the expectation over states is based on policy $\pi_C$, and $a_t, s_t, t \in \{1, 2, \ldots T\}$ are the observed actions and states.

Crucially, remote generation with a fixed reference distribution $Q$ does not reduce the communication rate in RRL compared to directly transmitting the actions with source coding. For example, if $Q$ is chosen as the uniform distribution over the action space, the expected rate is

$$\mathbf{D}_{KL}[P||U] = -\log|A| - \mathbf{H}(P) \text{ bits.}$$

As training progresses and the policy $P$ becomes more deterministic, $\mathbf{H}(P)$ decreases, and the cost approaches $-\log|A|$ bits—essentially the same as sending actions explicitly. This makes imitation learning a crucial component of GRASP.

## 5    Experiments

The two main claims of our work are that GRASP does not negatively impact training, and that it leads to significant communication savings. To evaluate its effectiveness, we assess it across a range of RL environments. We conduct experiments for both RRL paradigms: reward- and action-sending. For *reward-sending* schemes, where learning is localized to the actor, we consider communicating a full-precision 32-bit reward—dubbed full reward (FR)—as well as quantized reward (QR) to 16, 8, and 4 bits. For *action-sending* methods, where the controller learns the policy and dictates actions, we compare source coding of actions (called Action Source Coding (ASC)) with our proposed GRASP (based on remote generation). For single-agent settings, FR and ASC learn the same policy—the former at the actor, the latter at the controller—and thus produce identical returns (cumulative reward) during training. All the methods (including GRASP) are compatible with any RL algorithm. For our experiments, we focused on proximal policy optimization (PPO) (Schulman et al., 2017), a de-facto standard in RL. Additionally, we applied it to other algorithms such as deep Q-learning (DQN) (Mnih et al., 2013), soft Q-learning (SQ) (Haarnoja et al., 2017), and deep deterministic policy gradients (DDPG) (Lillicrap et al., 2016). We employ CleanRL open-source library (MIT license) implementation (Huang et al., 2022) using the default hyperparameters, if present, for each environment. These include neural network architecture, learning rate, and other algorithm-specific settings, with the full list provided in Appendix D. GRASP also entails learning the actor's policy in a behavioral cloning manner. For the actor, we utilize the same hyperparameters and architecture as the controller, training the policy using cross-entropy loss. For the implementation of remote generation, we opted for *ordered random coding* (Theis & Yosri, 2022) outlined in Appendix C. To ensure a comprehensive evaluation, we selected a diverse set of environments that vary in difficulty, type of action spaces (discrete and continuous), type of observations (fully and partially observable, proprioceptive, and image-based), and whether they involve single or multiple agents. These environments include CartPole and Pendulum from Classic Control, LunarLander and BipedalWalker from Box2D, HalfCheetah from MuJoCo, the Atari game Breakout, which were simulated using the Gymnasium library (Towers et al., 2023) (MIT license), as well as CooperativePong and PistonBall from the PettingZoo library (Terry et al., 2021) (MIT license and Apache license). The experiments were repeated across 20 independent and seeded runs, except for Breakout and CooperativePong, which were performed 8 times; all reported values are averaged and include the standard deviation. The experiments were performed on four Nvidia RTX 3080 GPUs with 10 GB of memory each, totaling 12 days of wall clock time including preliminary experiments; single runs for CartPole, Pendulum, LunarLander, and HalfCheetah took between 0.5 and 1.5 hours each, a BipedalWalker run 4 hours, while Breakout and CooperativePong runs 20 hours.

The single-agent training progress plots are presented in Figure 2. The first column describes the return throughout training; every 10 000 steps, the policy was evaluated across 30 episodes, recording the mean sum of

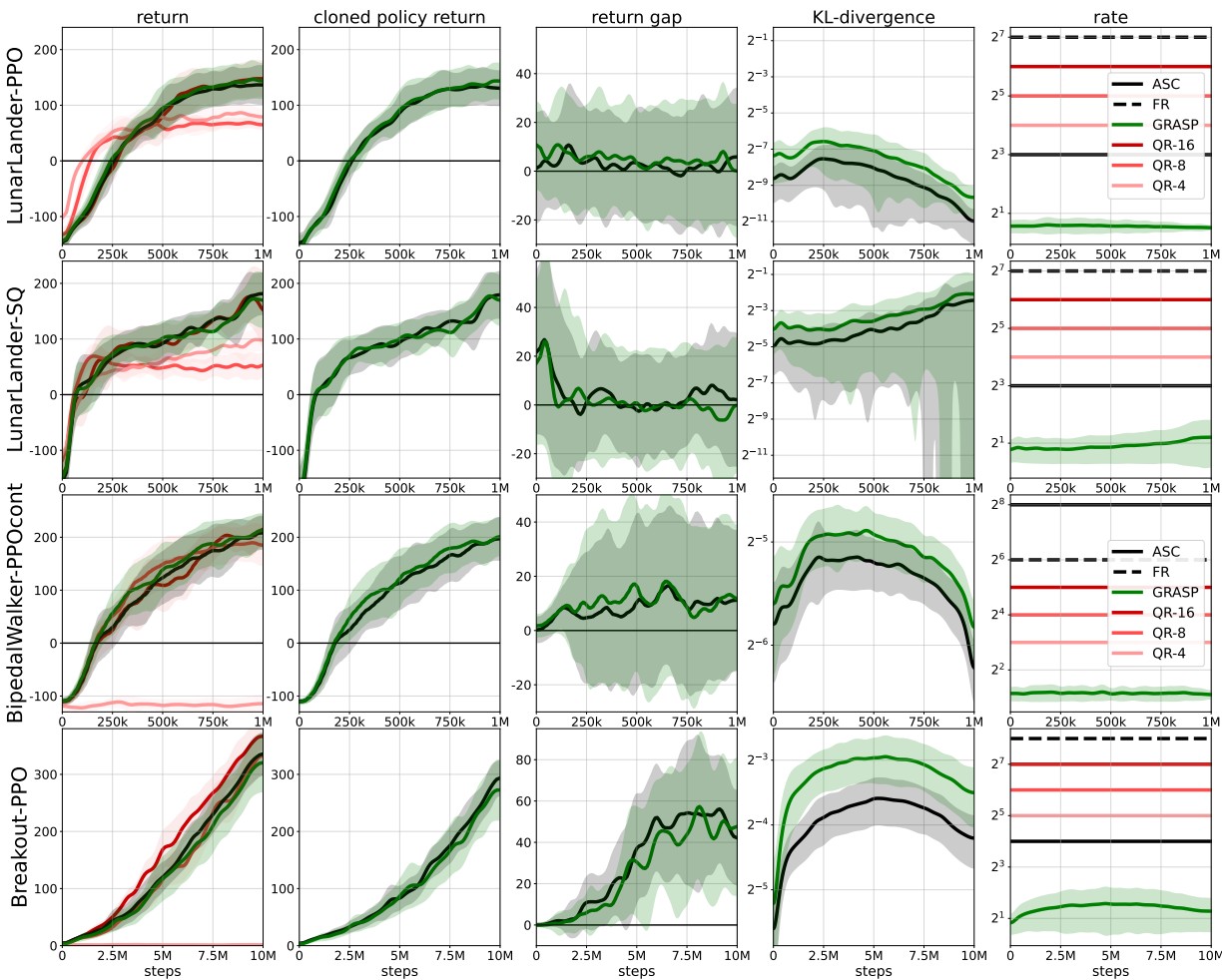

Figure 2: Training plots for different single-agent RL environments in the RRL setting. The plots compare action-sending methods—ASC and our proposed GRASP (action communication via remote generation combined with behavioral cloning)—against reward-sending schemes: full reward (FR), and quantized reward to 16, 8, and 4 bits (QR-16, QR-8, and QR-4, respectively). The algorithms used include PPO, continuous PPO, and Soft Q-Learning (SQL). Thick lines indicate the mean; shaded regions represent the standard deviation. For readability, values are smoothed with a Gaussian kernel (standard deviation: 2% of total training steps per environment).

rewards. Training performance is consistent among GRASP, ASC, FR, and even the 16-bit quantized reward (QR-16). Further reward quantization to 8-bit (QR-8) and 4-bit (QR-4) degrades the performance. Table 1 reports the controller's final returns (with standard deviations in parenthesis) for all tested environment-algorithm combinations, showing the same trend. In the table, the returns are normalized per environment: a score of 100 is assigned to the highest-return policy, and a score of 0 to a random policy. The second column in Figure 2 depicts the cloned policy's return for ASC and GRASP. Here, the cloned policy is the one obtained via supervised learning (behavioral cloning) by the actor from the controller's communicated actions (either directly as in ASC or via remote generation as in GRASP). As mentioned, this policy is not followed during training, but is used in remote action generation to reduce the communication cost in GRASP. In both cases, the training trajectories resemble the controller's policy, indicating the actor learns a useful policy via behavioral cloning. After training, depending on the use case, the controller might transmit its learned policy to the actor; alternatively, if the cloned policy is adequate, no further communication is needed.

Table 1: Returns in RRL normalized per environment

| Environment | Algorithm | GRASP | ASC | FR | QR-16 | QR-8 | QR-4 |
|---|---|---|---|---|---|---|---|
| CartPole | *PPO* | **100** (0) | **100** (0) | **100** (0) | **100** (0) | **100** (0) | -7 (3) |
| CartPole | *DQN* | **94** (9) | 81 (21) | 81 (21) | **89** (11) | **94** (18) | 50 (30) |
| CartPole | *SQ* | **93** (17) | **96** (11) | **96** (11) | **95** (10) | 82 (25) | 36 (10) |
| Pendulum | *PPOcont* | **100** (2) | **100** (2) | **100** (2) | **100** (1) | **100** (1) | -11 (9) |
| Pendulum | *DDPG* | **99** (2) | **99** (2) | **99** (2) | **99** (2) | **99** (2) | -16 (5) |
| LunarLander | *PPO* | **77** (7) | **76** (8) | **76** (8) | **79** (8) | 59 (2) | 62 (2) |
| LunarLander | *DQN* | **95** (8) | **100** (5) | **100** (5) | **99** (6) | **97** (9) | 84 (11) |
| LunarLander | *SQ* | **84** (11) | **87** (9) | **87** (9) | 78 (17) | 57 (4) | 67 (5) |
| BipedalWalker | *PPOcont* | **98** (9) | **96** (9) | **96** (9) | **100** (7) | 88 (11) | -1 (5) |
| HalfCheetah | *PPOcont* | **24** (5) | **24** (5) | **24** (5) | **22** (3) | **26** (8) | -4 (1) |
| HalfCheetah | *DDPG* | 82 (28) | **93** (27) | **93** (27) | **100** (17) | **92** (47) | -6 (4) |
| Breakout | *PPO* | 87 (13) | 92 (10) | 92 (10) | **100** (6) | 91 (9) | -0 (0) |
| CoopPong | *PPO* | 87 (5) | 90 (6) | 90 (6) | 92 (3) | 96 (3) | **100** (2) |
| Spread | *PPOcont* | **96** (12) | **100** (10) | **100** (10) | **95** (15) | 25 (23) | 67 (8) |
| PistonBall | *PPOcont* | **99** (3) | **100** (3) | **100** (3) | **99** (1) | **99** (2) | 91 (8) |

The actor's final policy performance is within a few percentage points of the controller's, demonstrating behavioral cloning as an effective alternative to policy transmission. The Breakout environment is an exception, explainable by undertraining, as its policy is still rapidly improving (and thus changing) even in the last steps of training. We can see the return gap in the third column of Figure 2 increasing at the beginning, then plateauing and starting to decrease towards the end of the training. Table 3 in Appendix A details the learned clone performance, validating this observation across all settings.

Figure 3 shows multi-agent training plots, following the same methodology. To test the approaches across more varied scenarios, our remote generation setup varies by environment: in CooperativePong, a single controller manages two agents acting independently, while their policies share weights. In PistonBall (20 actors) and Spread (3 actors), however, the controller is centralized, but each actor employs an independent, cloned policy for remote generation. The reward-sending baseline uses policies that share weights in all three scenarios. As before, performance is generally similar across these different approaches, degrading with more quantization—except in CooperativePong where, surprisingly, returns increase. In this instance, quantization appears to have acted as beneficial reward shaping, a technique which can also be employed with GRASP and ASC.

The communication costs for these alternatives are plotted in the final columns of Figures 2 and 3. For ASC, discrete action cost is the logarithm of action set cardinality; for continuous spaces, we followed the environments' specifications, which require 32-bit floats per action dimension. For GRASP, we used ordered random coding to communicate samples from the controller's policy, and calculated the log probability of the selected index as the communication cost. GRASP consistently outperforms all other algorithms, often by orders of magnitude. Table 2 outlines total communication costs: GRASP yields median 10.4-fold savings over ASC, averaging a 12.4-fold geometric reduction. Continuous action environments show the most significant savings. Sending the reward is functionally like ASC, differing mainly in that only the actor's model is trained; thus, intelligence—and computational complexity—resides at the actor. Assuming a 32-bit per time step communication rate, GRASP saves 6.3-fold to 343-fold in communication over sending the reward, with a 41-fold geometric average reduction. Compared to quantization, GRASP achieves average savings of 21-fold, 10-fold, and 5-fold for 16-bit, 8-bit, and 4-bit quantization, respectively. Crucially, only 16-bit quantization (QR-16) avoided performance degradation.

The communication cost of GRASP exhibits a consistent dynamic across most environments and learning algorithms. Initially, because the controller and actor networks are initialized identically via a common seed, their policies are perfectly aligned, resulting in a KL-divergence of zero and a near-zero communication

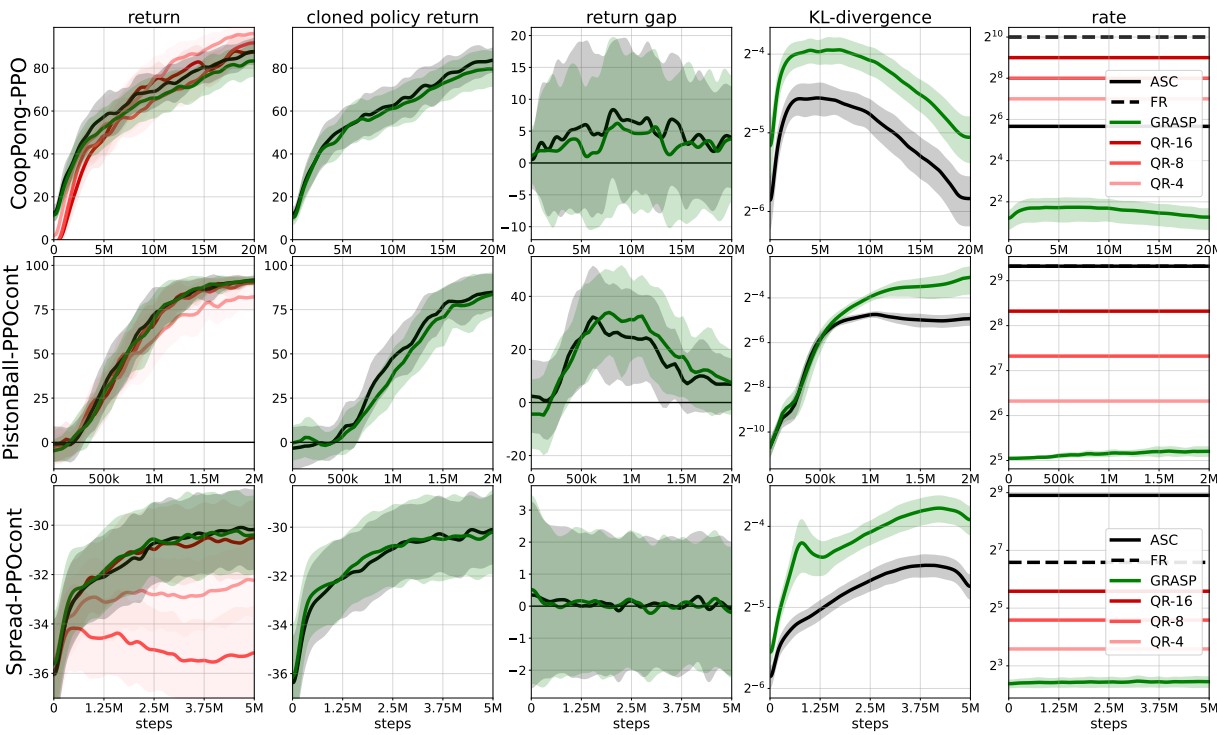

Figure 3: Training plots for different multi-agent RL environments in the RRL setting.

Table 2: Communication rate savings relative to FR baseline

| Environment | Algorithm | GRASP | ASC | FR | QR-16 | QR-8 | QR-4 |
|---|---|---|---|---|---|---|---|
| CartPole | *PPO* | ×**82.8** (1.68) | ×32.0 | ×1.0 | ×2.0 | ×4.0 | ×8.0 |
| CartPole | *DQN* | ×**51.2** (2.23) | ×32.0 | ×1.0 | ×2.0 | ×4.0 | ×8.0 |
| CartPole | *SQ* | ×**71.2** (3.29) | ×32.0 | ×1.0 | ×2.0 | ×4.0 | ×8.0 |
| Pendulum | *PPOcont* | ×**29.8** (0.39) | ×1.0 | ×1.0 | ×2.0 | ×4.0 | ×8.0 |
| Pendulum | *DDPG* | ×**10.4** (0.75) | ×1.0 | ×1.0 | ×2.0 | ×4.0 | ×8.0 |
| LunarLander | *PPO* | ×**86.5** (0.31) | ×16.0 | ×1.0 | ×2.0 | ×4.0 | ×8.0 |
| LunarLander | *DQN* | ×**36.1** (0.72) | ×16.0 | ×1.0 | ×2.0 | ×4.0 | ×8.0 |
| LunarLander | *SQ* | ×**67.4** (1.48) | ×16.0 | ×1.0 | ×2.0 | ×4.0 | ×8.0 |
| BipedalWalker | *PPOcont* | ×**28.7** (0.27) | ×0.2 | ×1.0 | ×2.0 | ×4.0 | ×8.0 |
| HalfCheetah | *PPOcont* | ×**43.0** (1.40) | ×0.2 | ×1.0 | ×2.0 | ×4.0 | ×8.0 |
| HalfCheetah | *DDPG* | ×6.3 (0.36) | ×0.2 | ×1.0 | ×2.0 | ×4.0 | ×**8.0** |
| Breakout | *PPO* | ×**95.2** (30.37) | ×16.0 | ×1.0 | ×2.0 | ×4.0 | ×8.0 |
| CoopPong | *PPO* | ×**343.5** (4.61) | ×20.2 | ×1.0 | ×2.0 | ×4.0 | ×8.0 |
| Spread | *PPOcont* | ×**17.7** (0.11) | ×0.2 | ×1.0 | ×2.0 | ×4.0 | ×8.0 |
| PistonBall | *PPOcont* | ×**18.1** (0.22) | ×1.0 | ×1.0 | ×2.0 | ×4.0 | ×8.0 |

cost. However, as the controller begins to learn from rewards, its policy evolves rapidly with each update. The actor's policy, which is trained via imitation, perpetually lags behind this moving target, causing the KL-divergence and the corresponding communication rate to rise. Finally, as training progresses and the controller's policy converges–stabilizing as it approaches an optimum and the learning rate anneals–the actor's policy can more accurately track the now-static target. This leads to a significant decrease in KL-divergence

and, consequently, a reduction in the communication cost. This reveals an efficient property of GRASP: it communicates only the necessary corrective information when the actor's policy is misaligned with the controller's.

## 6  Limitations

To perform remote generation, both parties require access to a common reference distribution $Q$. In GRASP, this is achieved by training an additional policy at the actor, which aims to follow the controller's policy as closely as possible. The closer the two policies are, the smaller the communication cost. This requirement introduces increased computational cost at the actor to reduce the communication rate. As previously mentioned, the need for a common distribution $Q$ can be circumvented by periodically transmitting the controller's current policy to the actor. This approach can reduce the need for training a separate policy at the actor, but it may lead to periodic spikes in communication load, depending on the frequency and size of the transmitted policy updates.

RRL assumes that both the agent and the controller have access to the same state/observation (or the controller's observation is superset of agent's). In situations where this is not the case, a common policy cannot be trained, and thus GRASP cannot be implemented. However, there exists a potential avenue due to recent advances in the information theory literature regarding the error rates of performing remote generation when the encoder and decoder do not share the same policies (Li & Anantharam, 2021). It remains to be determined how best to exploit different information available to the controller and the actor in such situations to find a good policy in a computation- and communication-efficient manner.

## 7  Conclusion

This work introduces RRL, a novel problem where a *controller*, exclusively observing the reward signal, guides remote *actors* through transmitted messages. There are two obvious benchmarks: In the first, the controller conveys the reward signal to the actors, so that the actors can learn the optimal policies by applying their favorite RL algorithm. In the second, the controller learns the optimal policy and transmits the optimal actions to the actors at each step. Both of these options may become infeasible when the agents need to coordinate their training or when the action set is prohibitively large or continuous. We have proposed a novel alternative method, called GRASP, based on importance sampling and behavioral cloning. The actor remotely generates a sample from the controller's policy, and to further reduce the communication cost, the actor attempts to estimate this policy through supervised learning. Our experiments show GRASP vastly outperforms these benchmarks by maintaining the same return while achieving 12-fold and 41-fold reductions in communication rate compared to action-sending and reward-sending alternatives, respectively.

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

# A  GRASP-ASC comparison

Table 3: Detailed performance of action-sending methods (GRASP vs. ASC) in RRL environments

| environment | algorithm | training method | controller final return | actor final return | return gap | norm. return gap (%) |
|---|---|---|---|---|---|---|
| CartPole | PPO | ASC | 500 (0) | 500 (0) | 0.0 (0.0) | 0.0 (0.0) |
| | | GRASP | 500 (0) | 500 (0) | 0.0 (0.0) | 0.0 (0.0) |
| CartPole | DQN | ASC | 415 (95) | 432 (81) | -16.7 (57.8) | -4.3 (14.7) |
| | | GRASP | 475 (40) | 458 (53) | 16.4 (43.1) | 3.6 (9.5) |
| CartPole | SQ | ASC | 481 (48) | 463 (68) | 17.4 (50.8) | 3.8 (11.1) |
| | | GRASP | 468 (77) | 463 (79) | 4.7 (14.6) | 1.1 (3.3) |
| Pendulum | PPOcont | ASC | -153 (20) | -154 (21) | 1.9 (4.8) | 0.2 (0.5) |
| | | GRASP | -153 (20) | -155 (22) | 1.9 (7.8) | 0.2 (0.7) |
| Pendulum | DDPG | ASC | -157 (28) | -246 (136) | 89.7 (126.1) | 7.4 (10.4) |
| | | GRASP | -156 (23) | -191 (81) | 35.4 (68.2) | 2.9 (5.6) |
| LunarLander | PPO | ASC | 135 (31) | 130 (27) | 5.1 (14.2) | 1.7 (4.7) |
| | | GRASP | 141 (29) | 142 (28) | -0.9 (16.9) | -0.3 (5.5) |
| LunarLander | DQN | ASC | 234 (22) | 207 (37) | 27.4 (29.2) | 6.6 (7.1) |
| | | GRASP | 215 (33) | 190 (30) | 25.2 (30.2) | 6.4 (7.7) |
| LunarLander | SQ | ASC | 180 (35) | 178 (44) | 1.3 (23.8) | 0.3 (6.2) |
| | | GRASP | 169 (44) | 169 (39) | -0.5 (21.4) | -0.1 (5.7) |
| BipedalWalker | PPOcont | ASC | 209 (28) | 196 (38) | 12.6 (17.8) | 3.9 (5.6) |
| | | GRASP | 214 (30) | 205 (33) | 9.6 (15.2) | 2.9 (4.7) |
| HalfCheetah | PPOcont | ASC | 1084 (251) | 1020 (233) | 63.4 (54.9) | 4.4 (3.8) |
| | | GRASP | 1058 (277) | 977 (253) | 81.4 (52.2) | 5.8 (3.7) |
| HalfCheetah | DDPG | ASC | 4662 (1429) | 3716 (1776) | 945.9 (1132.1) | 20.2 (24.2) |
| | | GRASP | 4113 (1449) | 3765 (1642) | 348.7 (766.1) | 8.4 (18.6) |
| Breakout | PPO | ASC | 340 (38) | 299 (29) | 41.4 (24.0) | 12.3 (7.1) |
| | | GRASP | 323 (49) | 274 (57) | 48.8 (29.4) | 15.2 (9.2) |
| CoopPong | PPO | ASC | 87 (6) | 85 (5) | 2.1 (6.2) | 2.6 (7.7) |
| | | GRASP | 84 (5) | 80 (4) | 3.9 (3.9) | 5.0 (5.1) |
| Spread | PPOcont | ASC | -30 (1) | -30 (1) | -0.1 (0.8) | -1.9 (12.2) |
| | | GRASP | -30 (1) | -30 (1) | -0.3 (0.8) | -4.4 (12.6) |
| PistonBall | PPOcont | ASC | 92 (3) | 85 (10) | 6.8 (9.5) | 7.3 (10.1) |
| | | GRASP | 91 (3) | 85 (11) | 5.8 (11.0) | 6.0 (11.4) |

# B  Additional results

In this appendix, we include other experiments mentioned in the main text. The training plots are depicted in Figure 4.

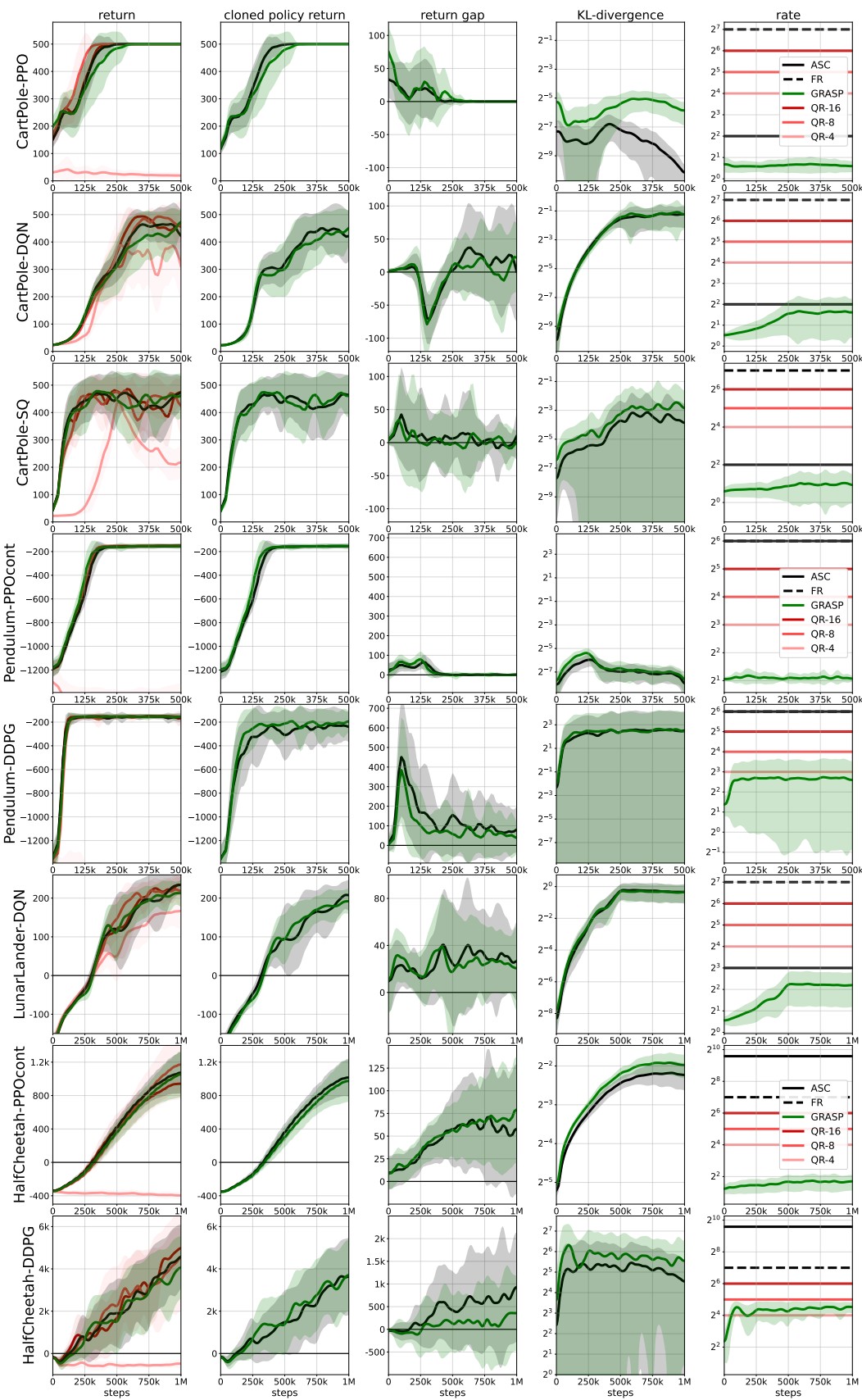

Figure 4: Supplementary training plots for RL environments in the RRL setting.

## C   Remote Generation

The remote generation method used throughout this work is *ordered random coding* from Theis & Yosri (2022), reproduced for convenience in Algorithm 3.

---
**Algorithm 3** Ordered Random Coding
---
**Require:** P, Q, N
1: $t, n, s^{\star} \leftarrow 0, 1, \infty$
2: $w = \min_x P(x)/Q(x)$
3: **repeat**
4:     $z \leftarrow$ sample $P$
5:     $v \leftarrow N/(N - n + 1)$
6:     $s \leftarrow t \cdot P(z)/Q(z)$
7:     **if** $s < s^{\star}$ **then**
8:         $s^{\star} \leftarrow s$
9:         $n^{\star} \leftarrow n$
10:    **end if**
11:    $n \leftarrow n + 1$
12: **until** $s^{\star} \leq t \cdot w$ **or** $n > N$
13: **return** $n^{\star}$

---

## D   Training and Hyperparameters

The experiments were performed on four Nvidia RTX 3080 GPUs with 10 GB of memory each, totaling 200 hours of wall clock time, including preliminary experiments. A single run of CartPole, Pendulum, LunarLander, and HalfCheetah took between 0.5 to 1.5 hours, BipedalWalker, Spread, and PistonBall took 4 to 6 hours, while Breakout and CooperativePong took 20 hours. The discount factor $\gamma$ was set to 0.99 for all environments. The hyperparameters for each of the experiments are presented in Tables 4, 5, 6, 7, and 8.

Table 4: Hyperparameter settings for PPO training

| env_id | total_timesteps | num_envs | learning_rate | num_steps | update_epochs | ent_coef | buffer_size | gae_lambda | clip_coef | vf_coef |
|---|---|---|---|---|---|---|---|---|---|---|
| CartPole-v1 | $5 \times 10^5$ | 4 | $2.5 \times 10^{-4}$ | 128 | 4 | 0.01 | $10^4$ | 0.95 | 0.2 | 0.5 |
| LunarLander-v2 | $10^6$ | 4 | $2.5 \times 10^{-4}$ | 128 | 4 | 0.01 | $10^4$ | 0.99 | 0.2 | 0.5 |
| BreakoutNoFrameskip-v4 | $10^7$ | 8 | $2.5 \times 10^{-4}$ | 128 | 4 | 0.01 | $10^4$ | 0.95 | 0.1 | 0.5 |
| cooperative_pong_v5 | $2 \times 10^7$ | 32 | $2.5 \times 10^{-4}$ | 128 | 4 | 0.01 | $10^4$ | 0.95 | 0.1 | 0.5 |

Table 5: Hyperparameter settings for PPOcont training

| env_id | total_timesteps | num_envs | learning_rate | num_steps | update_epochs | ent_coef | buffer_size | gae_lambda | clip_coef | vf_coef |
|---|---|---|---|---|---|---|---|---|---|---|
| Pendulum-v1 | $5\times10^5$ | 2 | $3\times10^{-4}$ | 2048 | 10 | 0 | $10^4$ | 0.95 | 0.2 | 0.5 |
| BipedalWalker-v3 | $10^6$ | 2 | $3\times10^{-4}$ | 2048 | 10 | 0 | $10^4$ | 0.95 | 0.2 | 0.5 |
| HalfCheetah-v4 | $10^6$ | 4 | $3\times10^{-4}$ | 2048 | 10 | 0 | $10^4$ | 0.95 | 0.2 | 0.5 |
| pistonball_v6 | $2\times10^6$ | 20 | $3\times10^{-4}$ | 2048 | 10 | 0 | $10^4$ | 0.95 | 0.1 | 0.1 |
| simple_spread_v2 | $5\times10^6$ | 3 | $3\times10^{-4}$ | 4096 | 10 | 0 | $10^4$ | 0.95 | 0.2 | 0.5 |

Table 6: Hyperparameter settings for DQN training

| env_id | total_timesteps | num_envs | learning_rate | num_steps | update_epochs | ent_coef | buffer_size | tau | start_e | end_e | explore_fract | learning_starts |
|---|---|---|---|---|---|---|---|---|---|---|---|---|
| CartPole-v1 | $5\times10^5$ | 4 | $2.5\times10^{-4}$ | 10 | 4 | 0.01 | $10^4$ | 1 | 1 | 0.05 | 0.5 | $10^4$ |
| LunarLander-v2 | $10^6$ | 4 | $2.5\times10^{-4}$ | 10 | 4 | 0.01 | $10^4$ | 1 | 1 | 0.05 | 0.5 | $10^4$ |

Table 7: Hyperparameter settings for SQ training

| env_id | total_timesteps | num_envs | learning_rate | num_steps | update_epochs | ent_coef | buffer_size | tau | start_e | end_e | explore_fract | learning_starts |
|---|---|---|---|---|---|---|---|---|---|---|---|---|
| CartPole-v1 | $5\times10^5$ | 4 | $2.5\times10^{-4}$ | 10 | 4 | 0.01 | $10^4$ | 1 | 1 | 0.05 | 0.5 | $10^4$ |
| LunarLander-v2 | $10^6$ | 4 | $2.5\times10^{-4}$ | 10 | 4 | 0.01 | $10^4$ | 1 | 1 | 0.25 | 0.9 | $10^4$ |

Table 8: Hyperparameter settings for DDPG training

| env_id | total_timesteps | num_envs | learning_rate | num_steps | update_epochs | ent_coef | buffer_size | tau | exploration_noise | learning_starts | noise_clip |
|---|---|---|---|---|---|---|---|---|---|---|---|
| Pendulum-v1 | $5 \times 10^5$ | 2 | $3 \times 10^{-4}$ | 1 | 2 | 0.01 | $10^6$ | 0.005 | 0.1 | 25000 | 0.5 |
| HalfCheetah-v4 | $10^6$ | 4 | $3 \times 10^{-4}$ | 1 | 3 | 0.01 | $10^6$ | 0.005 | 0.1 | 25000 | 0.5 |

