# OpenReview forum: "Remote Action Generation: Remote Control with Minimal Communication"
_TMLR — Rejected by TMLR_

### Review · Reviewer_5sL5 · 2025-11-11

**Summary Of Contributions:**

This paper introduces the problem of remote reinforcement learning with a communication constraint and proposes an algorithm (*guided remote action sampling* - GRASP) to solve it. This problem occurs in a multi-agent context, on which individual agents are provided only with partial information (no reward) and must interact with a controller to decide which action to carry out. Besides, there is an additional requirement that we want to reduce the communication between the controller and the agents. The main idea of the solving algorithm is to not directly communicate the rewards through the communication channel but instead to let the agents communicate a list of candidate actions to the controller. The controller will then use importance sampling to choose one action and then give back only the index of the desired action to the agents. Experiments are carried out on standard RL benchmarks (CartPole, Pendulum, etc.) and the results show that GRASP maintains the performances of while sending less information over the communication channel.

**Audience:**

No

**Audience Explanation:**

**Probably no.** The main issue I have with this paper is that it targets a very narrow and specific problem: multi-agent RL where the reward is only known by the controller, where few information must be common (state/observation/distribution Q), and where we want to minimize the bits sent over the communication channel. This is honestly acknowledged in the *limitations* section of the paper.

I do not consider these specific conditions to be a critical issue if there were at least a relevant case study that is addressed, but this aspect is completely missing. Related works only give high-level situations where the method has the potential to be useful, and the experiments only target standard (and easy) RL benchmarks. It is currently not clear on which settings one would want to use this algorithm. For such reasons, I am unsure that the contribution will be of interest for the TMLR community.

**Broader Impact Concerns:**

Nothing to report here

**Claims And Evidence:**

Yes

**Claims Explanation:**

**Mostly.** The methodology proposed to reduce the communication load seems sound, and the idea to communicate only the index of actions instead of full rewards is well motivated. Experiments, although they are only carried out on simple benchmark (see my next comment), are convincing in the sense that they allow to reduce the communication load while maintaining the performance over the full-reward baseline. On the negative side, formalization of the mathematical concepts can be improved (see my requested changed).

**Requested Changes:**

**Critical changes:**

-	Provide a relevant case study for this new problem and algorithm with comparisons with the same baselines.

**Minor changes:**

-	Consider changing the name of the algorithm. The acronym GRASP is already used for a well-known heuristic (*Greedy randomized adaptive search procedures*), which can introduce confusions.
-	Avoid Footnote for large explanation and prefer to integrate them in the text instead (page 5).
-	*Clarity of algorithm 1*: batch-size does not have the same font as the other variables.
-	*Algorithm 2*: it is not clear what “act” is doing on the algorithm (i.e. modification on the variable or on the system state). For instance, why there is no modifications on the state ?
-	Consider renaming $P$ by $P_\theta$ to make clear that it is a policy parametrized with $\theta$ (and prevent confusion with other parameters). Idem for $Q$.
- The encoding ($f^{(i)}$) and decoding  ($g^{(i)}$)  functions are not formally defined, and it is not clear what they are.

---

> ### Author Response · Authors · 2026-01-25
> **Response to Reviewer 5sL5**
>
> # Major changes:
> We believe that the current experiments sufficiently validate the paper's core claims without the inclusion of a new case study. Our evaluation covers high-dimensional image inputs (Atari), continuous control (MuJoCo), and multi-agent settings (PettingZoo), which are established community proxies for the systems described in our motivation. Crucially, these diverse environments demonstrate that GRASP achieves performance equivalent to standard RL methods while delivering significant savings in data transmission.
>
> # Minor changes:
> We agree with the minor changes suggested by the reviewer. In particular, we will:
> * Consider alternative names for the algorithm to avoid confusion with the existing heuristic.
> * Integrate the footnote into the main text.
> * Fix the font of the batch-size variable.
> * Clarify the notation in the GRASP algorithm and add the standard RL algorithm for context (observe, act, and learn). In general, “act in environment ($a_t$)” leads to different state in “$s_t \leftarrow$ observe the environment”.
> * Rename $P$ and $Q$ to $P_\theta$ and $Q_\phi$ where appropriate.
> * Explicitly explain the encoding $f$ and decoding $g$ functions in the remote generation appendix.

---

### Review · Reviewer_TgJD · 2025-11-25

**Summary Of Contributions:**

This paper formalizes the Remote Reinforcement Learning (RRL) problem, where a central controller observes rewards while remote actors do not, and communication is rate-limited. To address this setting, the authors propose GRASP (Guided Remote Action Sampling Policy), a communication-efficient algorithm that uses remote generation and behavioral cloning to allow the controller to guide each actor using only a few bits per timestep. GRASP enables multi-actor control with drastically reduced communication cost while matching the performance of standard RL.

**Audience:**

Yes

**Audience Explanation:**

The Remote Reinforcement Learning (RRL) is a less-studied, interesting setting, and can be relevant to real-world problems with larger-scale systems. This paper seems like an important step in this direction.

**Claims And Evidence:**

Yes

**Claims Explanation:**

The claims made in the submission are generally well supported by accurate and convincing evidence. The authors introduce the Remote Reinforcement Learning (RRL) setting and justify its relevance with concrete examples and formal definitions.

The proposed GRASP method is evaluated extensively across a list of RL environments spanning discrete, continuous, and multi-agent tasks, which provides a good empirical basis for the central claim that GRASP achieves drastically lower communication cost without degrading RL performance. (although that many of these environments are fairly simple environments, the multi-agent environments are interesting but do not seem very complex)

The communication-rate plots, ablations on KL divergence, and comparisons against reward-sending and action-sending baselines are presented clearly and consistently show the proposed advantages. The experiments align closely with the theoretical expectations of remote generation, and the results are reported transparently with normalized returns, bit-rate metrics, and comparisons across multiple RL algorithms. Overall, the evidence is good, and supports the paper’s major claims.

**Requested Changes:**

- The method relies on both controller and actors generating identical candidate action lists using shared randomness. This is essential for correctness but is explained only briefly. The multi-actor synchronization and independence of Q policies is a related thing. These can be further explained in the paper, ideally visually with some diagrams perhaps. That might help the reader better understand the idea.
- To my understanding, the Q for each actor can be different, as the controller can be sending different actions to different actors at each timestep, it is still a bit unclear to me will this be a problem in a larger-scale system operating over a longer period of time?
- Can have some more analysis on scalability and communication overhead with large numbers of actors. The examples given in the paper seem to be a bit simple/smaller scale.
- Curious if BC is used less frequently to keep updating the actors, how much will the performance drop? Additionally, is there way to update the actors more so that they are closer to the controller policy? (e.g. like maybe at some point the system will allow some more communication rate so further updating them is possible)

---

> ### Author Response · Authors · 2026-01-25
> **Response to Reviewer TgJD**
>
> ## 1. Common randomness, synchronisation and Q policies independence
> We will update the article to highlight the idea of a common list of samples between controller and agents, it guides the answers to each of the three questions. Common randomness is implemented via shared random seeds, ensuring the controller and actors generate identical candidate lists. In the multi-actor setting, each actor requires synchronisation with the central controller to enable the generation of a common list of samples but not with other agents. Finally, agent policy independence is not unique to GRASP but is the standard MARL formulation; each agent acts based on the state of the environment. To achieve coordination, some form of communication is required. However, conditioned on both state and communication the policies of agents are still independent and GRASP can be employed, as communication is just another action. We will clarify these three aspects in the revised manuscript.
>
> ## 2. Scaling
> In general, for n agents each policy can be different and the controller generates n actions. This is standard for centralised-learning decentralised execution (CLDE) multi-agent learning paradigm. The computational load of each agent is constant, while for the controller it grows linearly with the number of actors. The policy P can either be factored into n independent policies or be a joint distribution. In practice, for CLDE it is factored to enable decentralized execution; thus, differing Q distributions are not problematic. Similarly, communication complexity is constant for each agent and scales linearly for the centralized controller. We will explicitly detail these scaling considerations in the text.
>
> ## 3. Agent policy updates
> Crucially, the actors always execute actions sampled from the controller's policy P, regardless of their local proposal distribution Q. Thus, reducing BC frequency does not affect task performance or learning rates; it only increases communication cost due to higher KL-divergence. Even in the worst case where Q is static, this cost should be on the order of standard Action Source Coding. Regarding explicit updates, transmitting full policy weights to improve Q is inefficient, requiring an estimated 500x more bandwidth than our method (assuming ~100k parameters and infrequent updates every 1000 steps). The theoretical optimum for minimizing communication is maintaining a Bayesian belief over the controller's policy, our BC approach provides an efficient and computationally feasible approximation.

---

### Review · Reviewer_4Mky · 2026-02-25

**Summary Of Contributions:**

This paper deals with an RL setting where one central controller observes state and reward, but cannot act directly, while one or more actors observe state and can act but do not observe reward, and communication from controller to actors is rate-limited. The proposed method GRASP (Guided Remote Action Sampling Policy) aims to make the controller to actor communication rate-efficient. Instead of the controller transmitting a full action or transmitting rewards so actors can learn, GRASP uses an information-theoretic remote generation and communication of samples idea. Empirically, the paper reports that GRASP largely matches baseline training performance while substantially reducing estimated communication compared to sending rewards, and sending actions directly with a simple source-coding baseline, with particularly large gains in continuous-action domains.

**Audience:**

Yes

**Audience Explanation:**

This paper is positioned at a nice intersection of communication-efficient RL with distributed control, multi-agent training with centralized reward access, and information-theoretic communication of samples ideas that don’t show up often in mainstream RL papers. Even if some readers don’t care about the full RRL framing, the idea of converting send an action into send a tiny corrective index, while the actor learns a proposal distribution is a broadly reusable motif for bandwidth-constrained robotics, edge RL, and multi-agent coordination.

**Broader Impact Concerns:**

If actions are transmitted as compact indices relying on shared randomness, systems may become brittle to desynchronization or adversarial perturbations. For example, if an attacker can desynchronize the actor’s proposal policy, the controller’s messages may decode into unintended actions. A short discussion of failure modes and how to detect or resynchronize would strengthen the safety of the proposed method.

**Claims And Evidence:**

Yes

**Claims Explanation:**

Strengths
Multiple environments such as classic control, Box2D, MuJoCo, Atari, PettingZoo, and multiple RL algorithms such as PPO, DQN, soft Q-learning, DDPG are tested with multiple seeds and plotted learning curves plus summary tables.

Reported normalized returns show GRASP typically close to ASC and FR, and the training curves show broadly similar learning dynamics with predictable degradation for very low-bit reward quantization.

Tables and plots show GRASP’s rate dropping as imitation catches up, and the geometric-average fold reduction numbers are substantial, especially when the alternative is 32-bit floats per action dimension for continuous control.

Weaknesses
The reported rate seems computed from quantities like log-probability of an index or KL divergence, rather than from a concrete bitstream produced by a specified prefix code with overhead, framing, and resynchronization. For many research contexts that is acceptable as a theoretical proxy, but the paper’s claims are phrased as strong practical reductions.

Many realistic systems would use quantized actions of 8–16 bits, delta coding, or task-specific bounded precision. Since the paper already evaluates reward quantization at multiple bitrates, it would be natural to include action quantization baselines for continuous actions too.

GRASP’s construction relies on controller and actor sharing the reference distribution Q in lockstep without explicitly communicating network parameters. In practice, this requires either deterministic updates with shared randomness and identical numerics, or periodic resynchronization. Otherwise the decoder may not match the encoder’s assumed Q, and the scheme can fail catastrophically. This is partially mentioned as a limitation, but it needs more explicit treatment given how central it is.

**Requested Changes:**

As written, Algorithm 3 is not correct since it has degenerate initialization and unused variables. Please provide a correct, complete description of the remote generation procedure actually used in experiments, including what randomness is shared, how encoder and decoder remain synchronized, what parameter N means operationally and how it is chosen, whether the procedure is exact or approximate for continuous distributions, and what the approximation error is if any.

Several sentences in the background section read like the decoder is both selecting and receiving the index. Cleaning this up will prevent reader confusion about who samples what and who sends what.

State precisely how “rate” is computed in experiments. If no explicit entropy coding is implemented, say so and frame results as idealized bitrate estimates.

Add stronger baselines for continuous actions such as action quantization at 16/8/4 bits per dimension, and a periodic policy sync baseline, since you already discuss this alternative conceptually.

Discuss compute/latency tradeoffs explicitly. Provide a rough per-step computation or wall-clock overhead compared to ASC/FR. Show sensitive GRASP is to weaker actor models with smaller networks or fewer imitation updates. Show whether communication savings persist when actor compute is capped, which is a common real constraint for edge devices.

---

> ### Author Response · Authors · 2026-05-03
> **Response to Reviewer 4Mky**
>
> ## 1. Ordered Random Coding
> Algorithm 3 lacks an update for the variable t; we will fix this. In practice, to be able to sample the same realisations of random variables, the encoder and decoder simply share a common seed. We do not introduce any additional synchronisation mechanisms, since each message is uniquely decodable and the stream of bits is unambiguous. The parameter N is a feature of the particular channel simulation algorithm we used; it bounds the number of samples used but introduces bias. If \(N = 2^{D_{KL}(P \|\| Q) + t}\), then the total variation between the desired sample distribution P and the actual distribution falls exponentially as t increases. In practice, consistent with other works in the field, we found that t = 0 worked well. There also exist algorithms that produce exact outputs even for continuous distributions, such as the Poisson Functional Representation. We will update the paper to include these points.
>
> ## 2. How is the rate computed?
> We compute the communication cost as the negative log-probability of the message. We will explicitly clarify this in the revised manuscript.
>
> # Requested changes
>
> * **Add quantized actions and periodic sync baseline.**
>
> We will add action quantisation as an additional baseline. As outlined in our response to Reviewer 5sL5, a periodic sync baseline would require a 500-fold increase in the rate, making it uncompetitive.
>
> * **Provide wall clock overhead.**
>
> The wall-clock time of GRASP was approximately twice that of ASC. However, this might be due to our implementation of remote generation, which could be further optimised. We will test a modified setup where the edge device network is smaller to evaluate the benefits of the asymmetric setting.
>
> * **Clarifying encoder and decoder roles**
>
> We will clarify that the decoder receives the index of a sample and then outputs the corresponding realisation from the list.

---

### Review · Reviewer_P85F · 2026-03-07

**Summary Of Contributions:**

The paper first introduced a new RL problem called remote reinforcement learning (RRL) with communication constraints, where there is a single controller and multiple actors that communicate with the controller through a rate-limited channel. Then, the paper proposed a new RL method called GRASP, where the controller solves a standard RL problem, and actors employ importance-based sampling to generate actions from the distribution of the controller's policy. Experiment results demonstrate GRASP achieves robust performance across single-agent and multi-agent RL scenarios.

**Additional Comments:**

- While I appreciate the new formulation of the problem and it seems interesting, I think the experiment section could be more persuasive by conducting experiments in more realistic scenarios. As mentioned in the introduction channel, controlling smart grid control or traffic signal control are exactly real-world problem that can evaluate the effectiveness of the framework and there already exists several benchmarks that can test several RL methods [1,2].


[1] Nweye, Kingsley, et al. "The citylearn challenge 2022: Overview, results, and lessons learned." NeurIPS 2022 Competition Track (2023): 85-103.

[2] Ault, James, and Guni Sharon. "Reinforcement learning benchmarks for traffic signal control." Thirty-fifth Conference on Neural Information Processing Systems Datasets and Benchmarks Track (Round 1). 2021.

**Audience:**

Yes

**Audience Explanation:**

- In real-world scenarios, there may be a central controller, but each agent has limited capabilities of generating actions. The new formulation of RRL seems crucial and important at least for me.

**Claims And Evidence:**

Yes

**Claims Explanation:**

- The authors try to make a clear distinction between prior literature, such as MARL, asynchronous RL, or federated RL, and RRL by providing several example scenarios and diagrams.

- Experiment results on diverse tasks with diverse base RL algorithms support their claim.

**Requested Changes:**

- It seems like the tasks in the experiment have a very low-dimensional action. Could authors consider the case where the dimension of action is higher? (e.g., Humanoid)

- What if the message is corrupted with a certain probability? I think this scenario is also possible in remote reinforcement learning.

- What if there is a time constraint for generating actions? I think this scenario is also possible in remote reinforcement learning, and importance sampling with a time constraint may lead to wrong decisions.

---

> ### Author Response · Authors · 2026-05-03
> **Response to Reviewer P85F**
>
> ## 1. Higher-dimensional action spaces
> Our method scales to higher-dimensional action spaces, provided that the Kullback-Leibler (KL) divergence remains low. Under this condition, there is no theoretical reason why the approach would not succeed in more complex environments (such as Humanoid).
>
> ## 2. Noisy channels
> We find this to be an interesting direction. In the current work, we assume the channel is a reliable bit-pipe with no communication errors. However, incorporating channel noise and message corruption would be a valuable avenue for future research.
>
> ## 3. Time constraints
> On the agent side, the only overhead is the communication delay. Crucially, the agent does not need to generate the entire list of samples in advance; instead, it can wait to receive the index from the controller and generate only the sample at that position on demand (e.g., using a shared seed and the received index). On the controller side, the algorithm is highly parallelisable and can be accelerated accordingly. In particular, since the sampling algorithm requires finding a minimum over a list, a divide-and-conquer approach can be applied to scale this step massively.

---

### Decision · Action_Editor_YRVW · 2026-04-02

**Recommendation:** Reject

**Additional Comments:**

I recommend rejecting this paper at this point. The authors did not engage into a discussion with the latest 2 reviews. In addition, while the authors replied to the first two reviews, sadly they did not set the visibility of the responses such that the reviewers could actually see them. However, based on my own convictions, at least some of the answers are unsatisfactory.

I recommend rejection but giving the authors the option to consider to carefully revise the manuscript to address all the reviewers' concerns and then to submit a major revision at a later time.

**Audience:**

No

**Audience Explanation:**

While the proposed RRL formulation is interesting, it targets a very narrow and specific problem of multi-agent RL where the reward is only known by the controller. Also under the assumption that some information must be common (state/observation/distribution Q), and under the second assumption that the bits sent over a (narrow) communication channel should be minimized. The reviewer recommendations are not consistent and maybe the authors can re-frame the contribution to open it up for a wider audience.

**Claims And Evidence:**

No

**Claims Explanation:**

Partly. While the experiments in general give sufficient insight that the proposed method in general works, the experiment section should be more persuasive using experiments in more realistic scenarios. In particular the experiments should pick up the example from the introduction (channel, controlling smart grid control or traffic signal control). Also, there already exists several benchmarks that can test several RL methods [1,2].

[1] Nweye, Kingsley, et al. "The citylearn challenge 2022: Overview, results, and lessons learned." NeurIPS 2022 Competition Track (2023): 85-103.

[2] Ault, James, and Guni Sharon. "Reinforcement learning benchmarks for traffic signal control." Thirty-fifth Conference on Neural Information Processing Systems Datasets and Benchmarks Track (Round 1). 2021.

**Resubmission Of Major Revision:**

The authors may consider submitting a major revision at a later time.

---

> ### Author Response · Authors · 2026-05-03
> **Response to Action Editor YRVW**
>
> Thank you for the review and final assessment.
>
> We believe our experiments support our claims and that sharing remote RL is of interest to the TMLR community. The reviewers agreed with this assessment: 4 out of 4 concluded the evidence supports the claims, and 3 out of 4 agreed the findings would interest the audience.
>
> Regarding our engagement, our responses to the final two reviews were delayed because we were actively working on the requested experiments, and we will publish those responses now. Lastly, we believe all of our previous responses are currently visible to everyone on the platform and are not hidden from the reviewers.